

# Hydroclimatic control on suspended sediment dynamics of a regulated Alpine catchment: a conceptual approach

Anna Costa[1], Daniela Anghileri[1], Peter Molnar[1]

[1]Institute of Environmental Engineering, ETH Zurich, 8093 Zurich, Switzerland

*Correspondence to*: Anna Costa (costa@ifu.baug.ethz)

**Abstract.** We analyse the control of hydroclimatic factors on suspended sediment concentration (SSC) in Alpine catchments by differentiating among the potential contributions of erosion and suspended sediment transport driven by erosive rainfall, defined as liquid precipitation over snow free surfaces, icemelt from glacierized areas, and snowmelt on hillslopes. We account for the potential impact of hydropower by intercepting sediment fluxes originated in areas diverted to hydropower

reservoirs, and by considering the contribution of hydropower releases to SSC. We obtain the hydroclimatic variables from daily gridded datasets of precipitation and temperature, implementing a degree–day model to simulate spatially distributed snow accumulation and snow–ice melt. We estimate hydropower releases by a conceptual approach with a unique virtual reservoir regulated on the basis of a target–volume function, representing normal reservoir operating conditions throughout a hydrological year. An Iterative Input Selection algorithm is used to identify the variables with the highest predictive power

for SSC, their explained variance, and characteristic time lags. On this basis, we develop a hydroclimatic multivariate rating curve (HMRC) which accounts for the contributions of the most relevant hydroclimatic input variables mentioned above. We calibrate the HMRC with a gradient–based nonlinear optimization method and we compare its performance with a traditional discharge–based rating curve. We apply the approach in the upper Rhone Basin, a large Swiss Alpine catchment, heavily regulated by hydropower. Our results show that the three hydroclimatic processes – erosive rainfall, icemelt, and snowmelt –

are significant predictors of mean daily SSC, while hydropower release does not have a significant explanatory power for SSC. The characteristic time lags of the hydroclimatic variables correspond to the typical flow concentration times of the basin. Despite not including discharge, the HMRC performs better than the traditional rating curve in reproducing SSC seasonality, especially during validation at the daily scale. While erosive rainfall determines the daily variability of SSC and extremes, icemelt generates the highest SSC per unit of runoff, and represents the largest contribution to total suspended

sediment yield. Finally, we show that the HMRC is capable of simulating climate–driven changes in fine sediment dynamics in Alpine catchments. In fact, HMRC can reproduce the changes in SSC in the past 40 years in the Rhone Basin connected to air temperature rise, even though the simulated changes are more gradual than those observed. The approach presented is this paper, based on the analysis of the hydroclimatic control on suspended sediment concentration, allows the exploration of climate–driven changes in fine sediment dynamics in Alpine catchments. The approach can be applied to any Alpine

catchment with a pluvio–glacio–nival hydrological regime and adequate hydroclimatic datasets.



# 1 Introduction

Climate plays a dominant role in erosional and sediment transfer processes in Alpine catchments (e.g. Huggel et al., 2012; Micheletti and Lane, 2016; Palazón and Navas, 2016). In such environments, three main hydroclimatic forcings drive the processes that contribute to suspended sediment concentration (SSC) along channels: erosive rainfall, glacial melt, and
snowmelt. Erosive rainfall (ER), defined here as liquid precipitation over snow free surfaces, is responsible for soil detachment and erosion along hillslopes (Wischmeier, 1959; 1978), triggering of mass wasting events, such as debris flows and landslides (e.g. Caine, 1980; Dhakal and Sidle, 2004; Guzzetti, 2008; Leonarduzzi et al., 2017), which can mobilize large amounts of fine sediment (e.g. Korup et al., 2004; Bennett et al., 2012) resulting in very high suspended sediment concentrations in the receiving streams. Together with erosional processes along hillslopes, which are strongly related to
rainfall intensity (e.g. Van Dijk et al., 2002), precipitation events may also enhance channel and bank erosion through increased discharge. Icemelt (IM) is responsible for high concentrations of fine sediment produced with a variety of glacial erosion processes (Boulton, 1974). Icemelt may substantially increase suspended sediment concentration in glacially–fed streams by entraining and transporting fine sediment previously stored in subglacial networks and paraglacial environments (Aas and Bogen, 1988; Gurnell et al., 1996; Lawler et al., 1992). Snowmelt–driven overland flow (SM) generates hillslope
erosion and potentially affects channel and bank erosion by contributing to streamflow. This hydroclimatic forcing is important in Alpine environments where snowmelt can produce high hillslope runoff and be a major contributor to channel discharge (e.g. Grønsten and Lundekvam, 2006; Ollesch et al., 2006; Konz, 2012). Due to the diversity of the erosion and transport processes (e.g. erosion driven by overland flow, mass wasting events) and the variety of sediment sources involved (e.g. hillslopes, channels, glaciers), sediment fluxes generated by these three hydroclimatic variables are expected to
contribute to suspended sediment dynamics in a complementary way, both in terms of magnitude and timing.

In addition to natural hydroclimatic forcings, human activities potentially contribute to alter sediment dynamics, e.g. by changes in land use (e.g. Foster et al., 2003) and sediment storage in reservoirs (e.g. Syvitsky et al. 2005). In Alpine environments, it is especially water impoundment and flow regulation due to hydropower production which may substantially influence the suspended sediment regime (e.g. Anselmetti et al., 2007). The impacts of hydropower operations
on suspended sediment dynamics may vary substantially between catchments, depending on the specific features of the hydropower system (e.g. reservoir trapping efficiency, hydropower operations), and on the catchment characteristics (e.g. amount and grain size distribution of the eroded sediment, seasonal pattern of sediment production). Here, we focus on two main effects of hydropower operations: sediment trapping in reservoirs, and temporary sediment storage behind water diversion infrastructures (intakes), which may substantially reduce the amount of sediment delivered to downstream reaches
and/or significantly alter the timing of sediment release to the river network (e.g. Vörösmarty et al., 2003; Finger et al., 2006; Gabbud and Lane, 2016; Bakker et al., 2017). Despite sediment trapping, water released from hydropower reservoirs (HP) may carry suspended sediment either previously stored in the reservoirs or entrained along the downstream channels.





In the context of environmental change, it is important to understand how the sediment regime has changed and what the relative role of different hydroclimatic forcings may have been. There are examples of studies which demonstrated alterations in suspended sediment yields driven by changes in land use, climate, or by disturbances such as wildfires, earthquakes and flow impoundments (e.g. Loizeau and Dominik, 2000; Foster et al., 2003; Dadson et al., 2004; Yang et al.,

2007; Horowitz, 2010; Costa et al., 2017). These changes are normally addressed by calibrating different sediment rating curves models, which express suspended sediment concentration as a power function of discharge, for different sediment supply regimes and by making the parameters of the rating curves time dependent (e.g. Syvitski, 2000; Yang, 2007; Hu, 2011; Huang and Montgomery, 2013; Warrick, 2015). However these approaches do not explicitly address the sources of sediment and their activation by different hydroclimatic forcings and are limited to using discharge as a predictor. As a result

the hydroclimatic causality of changes in suspended sediment concentration in such analyses remains elusive. The approach proposed in this paper accounts explicitly for the hydroclimatic and hydropower activation/deactivation of different sediment sources, with the aim to identify their predictive power in estimating suspended concentration even without using discharge.

Our main objectives are: (1) to explore the role played by the hydroclimatic variables erosive rainfall ER, ice-melt IM, snow-melt SM, and the hydropower release HP, in controlling suspended sediment concentration of an Alpine catchment; and (2)

to analyse long–term, climate–driven changes in suspended sediment concentration on the basis of a conceptual, data–driven approach accounting separately for the contribution of ER, IM, SM and HP.

The upper Rhone Basin in southern Switzerland is used as the study catchment. The upper Rhone River contributes more than 65% of the total input of particulate matter into Lake Geneva, the largest lake in the Alps (Loizeau et al., 1997), substantially influencing the morphology and ecology of the river delta and the lake (Loizeau and Dominik, 2000; Loizeau et

al., 1997). The catchment is heavily regulated by hydropower infrastructure. Several large hydropower reservoirs have been in operation since 1960s, leading to a total retention capacity equal to roughly 20% of the total annual discharge of the catchment (Loizeau and Dominik, 2000; Fatichi et al., 2015). In addition to reservoirs, a complex network of water intakes and diversions extracts water from headwater streams and delivers it either to the major reservoirs or directly to the hydropower plants. From a detailed map of the hydropower scheme, including reservoirs and water diversions (Fatichi et al.,

2015), it is estimated that roughly 25% of the catchment is affected by hydropower: 8% flowing directly into the reservoirs, and 17% diverted through tunnels and pumping stations. Sediment fingerprinting conducted in the catchment in a recent study from Stutenbecker et al. (2017) indicates that sediment originated in the lithological unit more affected by hydropower is underrepresented at the outlet of the catchment, suggesting the impact of water impoundment on the sediment budget of the basin. In addition, alterations of suspended sediment concentration entering Lake Geneva have been observed in the

recent past and attributed to human impacts (Loizeau and Dominik, 2000; Loizeau et al., 1997) and changes in climatic conditions (Costa et al., 2017).

The paper is organized as follows: Sect. 2 describes the data pre–processing, the hydrological modelling procedure to obtain the hydroclimatic variables (ER, IM, SM), the approach to obtain the hydropower releases (HP), and the analysis performed to infer their link to suspended sediment concentration; Sect. 3 presents the upper Rhone basin and the data used in our




analysis; Sect. 4 reports the main results which are discussed in Sect. 5; and Sect. 6 concludes the manuscript by summarizing the main findings.

## 2 Methods

To analyse the role of hydroclimate on the suspended sediment regime of a catchment regulated by hydropower reservoirs, we first divide the catchment into two distinct areas: (1) the area which contributes to the runoff accumulated in hydropower reservoirs (regulated area), including the fraction of the catchment draining directly into the reservoirs and the fraction connected to the reservoirs through tunnels and pumping stations, and (2) the remaining area, which naturally flows to the river network (unregulated area).We assume that the sediment fluxes originated in the unregulated area contribute directly to SSC at the outlet of the catchment, while sediment fluxes generated in the regulated area are diverted into the reservoirs and later totally or partially released according to hydropower operations. Finally, we estimate the contribution to SSC at the outlet of the catchment of sediment fluxes originated in the unregulated area by ER, IM and SM, and of sediment fluxes carried by water released from the reservoirs during hydropower operations HP.

Our methodology consists of four main steps: (1) the derivation of mean daily $SSC_t$ and $ER_t$, $SM_t$, $IM_t$, $HP_t$ datasets – mean daily $SSC_t$ at the outlet of the catchment is derived from continuous measurements of turbidity (Sect. 2.1), the hydroclimatic input variables mean daily $ER_t$, $SM_t$, $IM_t$ are derived from spatially distributed snow and icemelt models, and the mean daily water releases from hydropower reservoirs $HP_t$ is derived by a conceptual approach based on a unique virtual reservoir, which is intended to model the cumulative effect of multiple reservoirs, when present in the catchment, and a target volume function (Sect. 2.2); (2) we use an Input Variable Selection algorithm to identify the variables with the highest predictive power for $SSC_t$ and we estimate their characteristic time lags (Sect. 2.3); (3) we calibrate and validate a rating curve accounting for the variables identified in the previous step (Hydroclimatic Multivariate Rating Curve – HMRC), and we evaluate the contribution of each hydroclimatic and hydropower component to $SSC_t$ (Sect. 2.4); (4) we apply the HMRC to simulate 40–year long time series of $SSC_t$ at the outlet of the catchment to investigate the impact of changes in climatic conditions on suspended sediment dynamics, and we compare simulated values with observations obtained with a traditional rating curve (RC) based on discharge only (Sect. 2.5).

### 2.1 Estimate of daily suspended sediment concentration

The specific operations described in this and the following paragraphs strongly depend on the data availability for the case study under consideration. In the following, we describe the operations we carried on for the upper Rhone Basin, but we also comment about the applicability of these and alternative operations to other catchments.

SSC sampling has been historically conducted manually, usually with low frequency (e.g. a few samples a week) and fixed intervals, because manual measurements are costly and time consuming (e.g. Gippel, 1995; Pavanelli and Pagliarani, 2002). This results in long but intermittent SSC datasets, which are not suitable for data–driven modelling, because they might not





be representative of the entire range of possible suspended sediment concentrations. On the other hand, automatic gauging stations with optical turbidity sensors produce turbidity datasets which are continuous but usually shorter, because of the recent wide–spread availability and installation of such sensors. Because turbidity is strongly related to suspended sediment concentration (e.g. Gippel, 1995; Lewis, 1996; Pavanelli and Pagliarani, 2002; Holliday et al., 2003; Lacour et al., 2009;

Métadier and Bertrand–Krajewski, 2012), the two datasets, when available at the same location, can be combined to obtain a high frequency SSC dataset. In our case, punctual manual measurements of SSC are collected twice per week at the outlet of the Rhone Basin and continuous measurements of Nephelometric Turbidity Units (NTU) are available for an overlapping period in 2013–2017 at the same location. To build a SSC–NTU relationship, we consider simultaneous measurements of NTU and SSC (i.e. with a maximum time lag of five minutes), after removing observations greater than the 90[th] percentile

(corresponding to 2000 mg l$^{-1}$ and 1000 NTU respectively) because we are concerned about errors in observed high sediment concentration pulses due to the punctual bottle–sampling procedure and known measurement errors at high NTUs, and the fact that SSC and NTU measurements are not taken exactly at the same location in space (and time) in the cross–section. We use least squares regression to fit the model after a logarithmic transformation of the variables:

$$SSC = a_0 \cdot NTU^{b_0} \tag{1}$$

For the back–transformation from the logarithmic to the linear scale, we applied the correction factor proposed by Duan (1983). Finally, we compute mean daily NTU values from continuous measurements of turbidity, and we use the SSC–NTU relation (Eq. 1) to estimate mean daily SSC.

## 2.2 Hydroclimatic Data Modelling

Datasets of the hydroclimatic variables ER, IM and SM need to be derived by hydrological modelling. The choice of the
model should be driven by the data availability for calibration and the required accuracy of the simulated outputs. In our case, we use a conceptual and spatially distributed model of snow and icemelt driven by spatially distributed precipitation and temperature (Costa et al., 2017). We use gridded datasets of mean daily precipitation and mean, maximum, and minimum daily air temperature to divide precipitation into rainfall and snowfall on the basis of a temperature threshold. We model ice and snow accumulation and melting with a degree–day approach (e.g. Hock, 2003). Icemelt occurs only on glacier
cells that are snow–free. Likewise, erosive rainfall occurs only on snow–free hillslope cells. We set temperature thresholds for snow/rain division (1°C) and for snow and icemelt initiation (0°C) based on the literature and on previous studies (e.g. Fatichi et al., 2015; Costa et al., 2017), while we calibrate melt factors with satellite–derived snow cover (MODIS) and with discharge measured at different locations in the catchment. We first calibrate the snowmelt rate from snow cover maps by spatial statistics that measure the grid–to–grid matching of the model. Second, we calibrate the icemelt rate on the basis of
discharge measured at the outlet of two highly glaciated sub–catchments. For more details on the hydrological model description and calibration see Costa et al. (2017). Finally, we sum the spatially distributed hydroclimatic variables over the regulated and unregulated areas and we obtain respectively mean daily $ER_t^{HP}$, $IM_t^{HP}$, $SM_t^{HP}$ and $ER_t$, $SM_t$, and $IM_t$.





We represent all the hydropower reservoirs operating in the catchment with a unique virtual reservoir, because data of water releases from individual reservoirs are seldom available. The release from the virtual reservoir is estimated on the basis of a target–volume function which represents the reservoir operations in normal conditions. For each day of the year, the hydropower release from the virtual reservoir $HP_{t+1}$ within the interval from day t to day t+1 is estimated as the difference of

the reservoir storage and the target volume, when positive, zero otherwise. The reservoir storage $V_{t+1}$ is finally computed on the basis of the mass balance:

$$V_{t+1} = V_t + I_{t+1} - HP_{t+1} \qquad (2)$$

where $I_{t+1}$ represents the inflow into the virtual reservoir within the interval from day t to day t+1.

To derive the capacity of the virtual reservoir, we consider the 13 largest reservoirs operating in the Rhone catchment. A list

of the reservoirs with their retention capacity is reported in Table S1 of the Supplementary Material, while their spatial location is shown in Figure 1. We compute the target–volume functions of each individual reservoir by averaging observed storage time series for reservoirs when observations are available and by adopting normalized reference curves within the individual reservoir regulation range otherwise (see Fatichi et al., 2015 for the full details). We then compute the target– volume function of the virtual reservoir by adding the target–volume functions of each individual reservoir and by scaling

the sum to the total annual inflow. We compute the daily inflow $I_t$ in Eq. (2) as the sum of the three hydroclimatic fluxes, erosive rainfall, icemelt and snowmelt generated over the regulated area:

$$I_t = ER_t^{HP} + IM_t^{HP} + SM_t^{HP} . \qquad (3)$$

It has to be noted that $I_t$ represents direct potential runoff from the regulated area without accounting for evapotranspiration and infiltration losses. We therefore scale the capacity of the virtual reservoir, to obtain reservoir seasonal dynamic

resembling the available observations. For the scaling, we assume the minimum volume of the virtual reservoir equal to zero, and the maximum volume equal to 70% of the total annual inflow into the reservoirs, which roughly corresponds to the average ratio between storage capacity and total annual inflow. The procedure described above implies that all reservoirs of the catchment are regulated following the same operational rule driven by the seasonality of inflow, i.e. an annual cycle of drawdown during winter and refill during spring and summer. Due to their geographical proximity, the similar elevation, and

the available observations, this assumption can be considered realistic. We validate the hydropower operations model by comparing the mean daily normalized values of simulated hydropower releases of the virtual reservoir and observations from Mattmark, a reservoir with a volume capacity of $10^8$ m$^3$ located in the upper part of the catchment. Although our hydropower operations model is relatively simple, the comparison shows a good agreement with the observations (Fig. S1 of the Supplementary Material).

**2.3 Input Variable Selection Algorithm**

We apply the Iterative Input Selection (IIS) algorithm (Galelli and Castelletti, 2013) to (1) select which variables play a significant role in predicting $SSC_t$, (2) quantify their relative importance, and (3) identify the time lags of the sediment flux associated with each selected variable. The IIS algorithm selects the most relevant input variables, among a set of candidate



input variables (in our case mean daily $ER_{t-l}$, $SM_{t-l}$, $IM_{t-l}$, $HP_{t-l}$ at different time lags l), to predict a specific output variable (in our case mean daily $SSC_t$). It calibrates and validates a series of regression models considering different sets of input variables and selecting the ones that display the best model performances. The algorithm adopts Extremely Randomized Trees, or Extra–Trees, (Geurts et al., 2006) as regression models, because they allow dealing with non–linear

relations between input and output variables in a computationally efficient way. The Extra–Trees regression is based on a recursive splitting procedure, which partitions the dataset into sub–samples containing a specified number of elements. This splitting procedure is performed several times by randomizing both the input variable and the cut–point used to split the sample, in order to minimize the bias of the final regression (for more details see Geurts et al., 2006).

The IIS algorithm is based on an iterative procedure, which allows for the ranking of the candidate input variables according

to their significance in explaining the output variable on the basis of the coefficient of determination $R^2$ of the underlying regression model. At the first iteration, regression models are identified and the candidate variable leading to the best model performance is selected. At subsequent iterations, the original output variable (i.e. $SSC_t$) is substituted with the residual of the model computed at the previous iteration to minimize the redundancy related to the selection of input variables that are highly correlated between each other (Galelli and Castelletti, 2013). Because of the relatively short duration of our dataset

and the marked seasonal pattern that characterizes the considered candidate input variables and output variable, we randomly shuffle the dataset 100 times before running the IIS algorithm as suggested by Galelli and Castelletti (2013). Among the 100 runs of the algorithm, we choose the most frequently selected model and the most frequently selected model including hydropower releases, if the two do not correspond. We then analyse the selected input variables, their characteristic time lags and the fraction of variance explained by each selected variable.

**2.4  Relative contribution of hydroclimatic forcing to SSC**

To further investigate the contribution of hydroclimatic forcing to suspended sediment dynamics, we propose a non–linear multivariate rating curve (Hydroclimate Multivariate Rating Curve, HMRC), which relates $SSC_t$ to the hydroclimatic variables described above, representing the main drivers for the suspended sediment regime of an Alpine catchment:

$$SSC_t = a_1 \cdot ER_{t-l_1}^{b_1} + a_2 \cdot IM_{t-l_2}^{b_2} + a_3 \cdot SM_{t-l_3}^{b_3} + a_4 \cdot HP_{t-l_4}^{b_4}, \tag{4}$$

where $ER_{t-l_1}$, $IM_{t-l_2}$, $SM_{t-l_3}$ are mean daily erosive rainfall, icemelt, and snowmeltover unregulated areas, computed at time $t-l_1$, $t-l_2$, and $t-l_3$ respectively, and $HP_{t-l_4}$ is the daily release of water from the virtual hydropower reservoir at time $t-l_4$. $SSC_t$ is expressed in dg $l^{-1}$ while $ER_{t-l_1}$, $IM_{t-l_2}$, $SM_{t-l_3}$ and $HP_{t-l_4}$ are expressed as mean values over the catchment in mm day$^{-1}$. The time lags, $l_1$, $l_2$, $l_3$ and $l_4$, identified with the Input Variable Selection Algorithm (Sect. 2.3), represent the time necessary for sediment produced at a given location in the catchment to reach the outlet. In principle, the

travel time depends on the sediment source location (i.e. distance from the outlet) and the velocity of the transport (which is a function of runoff, topography, and flow resistance). Here, we assume a characteristic travel time for each hydroclimatic/hydropower component, i.e. $l_i$ (with i = 1, 2, 3, 4), which represents an average travel time in space (i.e., over





the catchment) and time (i.e., over the hydrological year). We also assume that coefficients $a_i$ and $b_i$ (with i = 1, 2, 3, 4) may vary between the hydroclimatic/hydropower variables, because they express sediment availability as well as the nonlinearity of $SSC_t$ production by each variable. The HMRC does not use discharge in the estimation of $SSC_t$.

We calibrate the parameters of the non–linear multivariate HMRC $a_i$ and $b_i$ in Eq. 4, by minimizing the mean squared error
(MSE) between observed and simulated SSC with a gradient–based optimization approach. We assume that each sediment flux originates under supply–unlimited conditions, i.e. there is a positive relation between sediment transport capacity and the load of sediment mobilized and transported. Accordingly, the optimization is subject to the following constraints: $b_i > -1$ (with i = 1, 2, 3, 4); coefficients $a_i$, (with i = 1, 2, 3, 4) are instead not constrained, which allows for dilution when $a_i < 0$; and simulated $SSC_t \geq 0$. We repeat the optimization procedure 100 times, starting from randomly generated
initial values to reduce the risk of detecting sub–optimal parameter configurations.

We evaluate the ability of the HMRC in reproducing mean daily $SSC_t$ time series observed at the outlet of upper Rhone Basin, and we compare its performance with a traditional rating curve (RC) which relates suspended sediment concentration to mean daily discharge $Q_t$ only:

$$SSC_t = a_{RC} \cdot Q_t^{b_{RC}} \tag{5}$$

We calibrate the parameters of the RC (Eq. 5) $a_{RC}$ and $b_{RC}$ by least squares regression applied to the logarithm of $SSC_t$ and $Q_t$. As for the SSC–NTU relation, we apply the smearing estimator of Duan (1973) to the back–transformed values of $SSC_t$ to correct for the bias (e.g. De Girolamo et al., 2015). The performance of the HMRC and RC models are evaluated by computing goodness of fit measures such as coefficient of determination $R^2$, Nash–Sutcliffe efficiency NSE, and root mean squared error RMSE, over the calibration and validation periods. We compare the simulated and observed seasonal patterns
of $SSC_t$ by analysing mean monthly values.

## 2.5 Long−term changes in SSC

Simultaneously with an abrupt rise in air temperature, the upper Rhone basin has experienced a statistically significant jump in mean annual SSC in mid–1980s, which has been attributed to an increase of icemelt and rainfall over snow free surfaces (Costa et al., 2017). To analyze the impact of changing climatic conditions on the long–term dynamics of suspended
sediment, we apply the rating curve based on hydroclimatic variables, HMRC, to simulate the time series of mean daily $SSC_t$ at the outlet of the upper Rhone basin for the 40–year period 1975–2015. We compare HMRC simulations both to the twice–a–week observations of SSC and to the values simulated with the traditional RC. We compare the three time series (observed, simulated with HMRC and with traditional RC) on the basis of mean annual values, computed by considering only simulations corresponding to SSC measurement days to allow for a fair comparison with observations. We apply
statistical tests for equality of the means on time series of mean annual SSC, simulated with the HMRC and the traditional RC, to test if the models can reproduce the shift of SSC detected in the observations.



## 3 Upper Rhone Basin: description and data availability

We apply our approach to the upper Rhone Basin in the Swiss Alps (Fig. 1). The total drainage area of the catchment is equal to 5 338 km$^2$ and about 10% of the surface is covered by glaciers. The topography of the basin which has been heavily preconditioned by uplift and glaciations (Stutenbecker et al., 2016) is characterized by a wide elevation range (from 372 to 4

634 m a.s.l.). The Rhone River originates at the Rhone Glacier and flows for roughly 170 km before entering Lake Geneva. The hydrological regime of the catchment is dominated by snow and ice melt with peak flows in summer and low flows in winter. Mean discharge is equal to about 320 m$^3$s$^{-1}$ in summer and 120 m$^3$s$^{-1}$ in winter, while the mean annual discharge is around 180 m$^3$s$^{-1}$. Basin–wide mean annual precipitation is about 1 400 mm yr$^{-1}$ and mean annual temperature is about 1.4 °C estimated at basin mean elevation.

Porte–du–Scex is the measurement station at the outlet of the Rhone River into Lake Geneva (Fig. 1), where the Swiss Federal Office of the Environment (FOEN) collects discharge, SSC and turbidity data. Mean daily discharge is available since 1905, while SSC is measured twice per week since October 1964. Quality‒checked continuous measurements of NTU are available since May 2013 (Grasso et al., 2012). SSC at the outlet is characterized by a seasonal pattern typical of Alpine catchments (Fig. 2a). During winter (December–March) sediment sources are limited because a large fraction of the

catchment is covered by snow and precipitation occurs in solid form. Streamflow is mainly determined by baseflow and hydropower releases (Loizeau and Dominik, 2000; Fatichi et al., 2015), and SSC assumes its minimum values. In spring, SSC increases when snowmelt–driven–runoff mobilizes sediments along hillslopes and in channels. Simultaneously, snow cover decreases and rainfall events over gradually increasing snow free surfaces erode and transport sediment downstream, resulting in SSC peaks. In July, SSC reaches its highest values in conjunction with streamflow (Fig. 2a). In late summer

(August and September), when icemelt dominates, sediment rich fluxes coming from proglacial areas maintain high values of SSC although discharge is decreasing (Fig. 2a). In terms of suspended sediment yield, low SSC conditions do not play a relevant role compared to moderate and high SSC conditions: more than 66% of the total suspended sediment load entering Lake Geneva during the 4–year period May 2013 – April 2017 is estimated to be due to SSC values greater than the 90th percentile (Fig. 2b).

The linear relationship between the logarithm of NTU and SSC for the overlapping period of measurement is statistically significant with a coefficient of determination $R^2 = 0.94$ (Fig. 3). After applying the correction factor for back–transforming from logarithmic to linear scale, the calibrated parameters of the relation in Eq. (1) are $a_0 = 0.56$ and $b_0 = 1.25$. This relation was used to convert NTU observations to mean daily SSC. We are aware that the relation between SSC and turbidity: (1) is site–specific, (2) may vary seasonally as function of discharge and transported grain sizes, and (3) depends on sediment

sources, because the size, the shape, and the composition of suspended material may influence values of turbidity (Gippel, 1995). For this reason, in this analysis: (1) we apply a site–specific SSC–NTU relation, (2) we calibrate the relation over a wide range of NTUs and discharge conditions to account for the seasonal variability in grain sizes transported by the flow, and (3) we derive the SSC–NTU relation based on a relatively short period of time, in which there is not evidence of changes





in sediment sources. In addition, by allowing a non–linear relation between SSC and NTU, we take partially into account the variability of turbidity with grain size. Higher suspended sediment concentrations are expected to transport proportionally larger grains, and the exponent in the SSC–NTU relation was expected to be greater than 1.

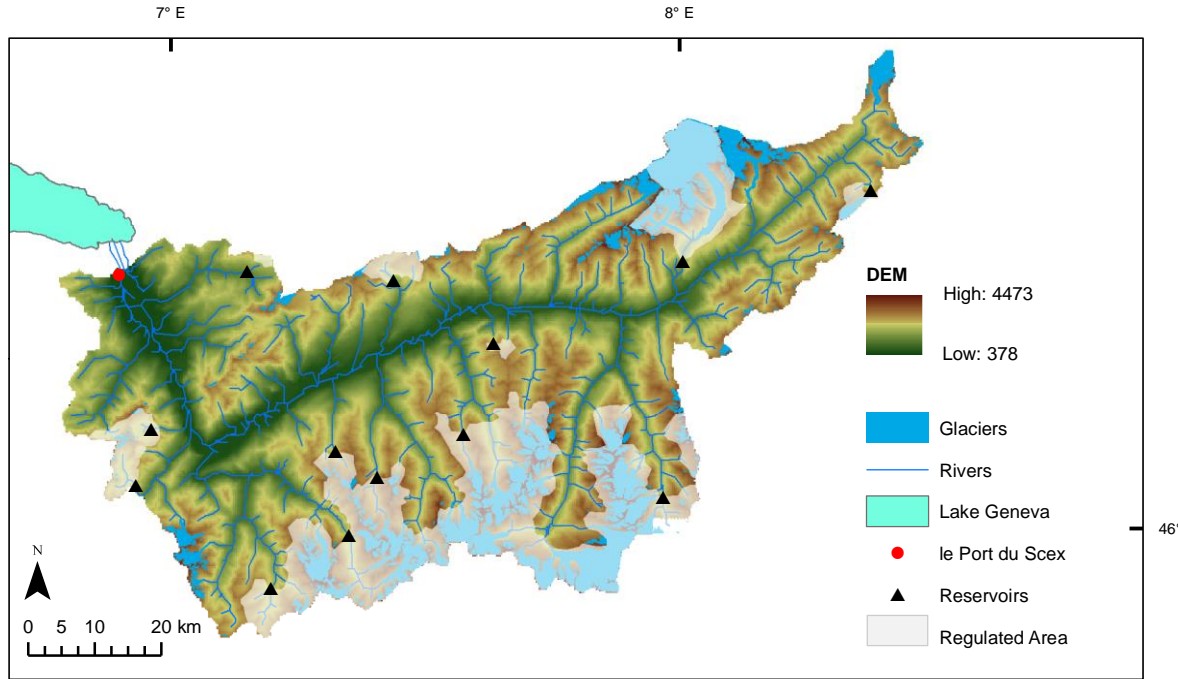

**Figure 1: Map of the upper Rhone Basin with topography, glacierized areas and river network. The measurement station le Port–du–Scex, located just upstream the Rhone River enters the Lake Geneva, is indicated with a red marker. The main 13 reservoirs considered in this study are represented with black triangles and the regulated fraction of the catchment is highlighted with light grey shaded area.**

We estimate the hydroclimatic variables for the forty year period 1975–2015 with the spatially distributed degree–day model of snow and icemelt. The model is implemented using a DEM with a spatial resolution of 250×250 m (Federal Office of Topography – Swisstopo). For the climatic dataset, we use gridded mean daily precipitation, mean, maximum and minimum daily air temperature at ~2×2 km resolution provided by the Swiss Federal Office of Meteorology and Climatology

15 (MeteoSwiss). These datasets are produced by spatial interpolation of quality–checked measurements collected at meteorological stations (Frei et al., 2006; Frei, 2014). Snow cover maps used for the calibration of the snowmelt rate were derived for the period 2000–2008 in a previous study (Fatichi et al., 2015) from the 8–day snow cover product MOD10A2 retrieved from the Moderate Resolution Imaging Spectroradiometer (MODIS) (Dedieu et al., 2010). We consider the GLIMS Glacier Database of 1991 to define the initial configuration of the ice covered cells. To calibrate the icemelt rate, we use



mean daily discharge data measured at the outlet of two highly glacierized tributary catchments: the Massa and the Lonza (Costa et al., 2017).

To separate sediment fluxes originated in regulated and unregulated areas of the catchment (Fig. 1), we used a detailed map of the main hydropower reservoirs and water uptakes and diversions, available from previous work of Fatichi et al. (2015)

and based on information included in the product ''Restwasserkarte'' available from the Swiss Federal Office for the Environment (BAFU).

When applying the IIS algorithm (Sect. 2.3), we consider mean daily $ER_{t-1}$, $SM_{t-1}$, $IM_{t-1}$ and $HP_{t-1}$ at time lags l from 0 day to 7 days. This choice is driven by the size of the basin and the expected flow concentration times in the basin. We calibrate the HMRC on data from the period 1 May 2013 – 30 April 2015 (730 days) and validate it over the period 1 May 2015 – 30

April 2017 (731 days). For the sake of comparison, calibration and validation periods are the same also when considering the RC.

(a)                                                                         (b)

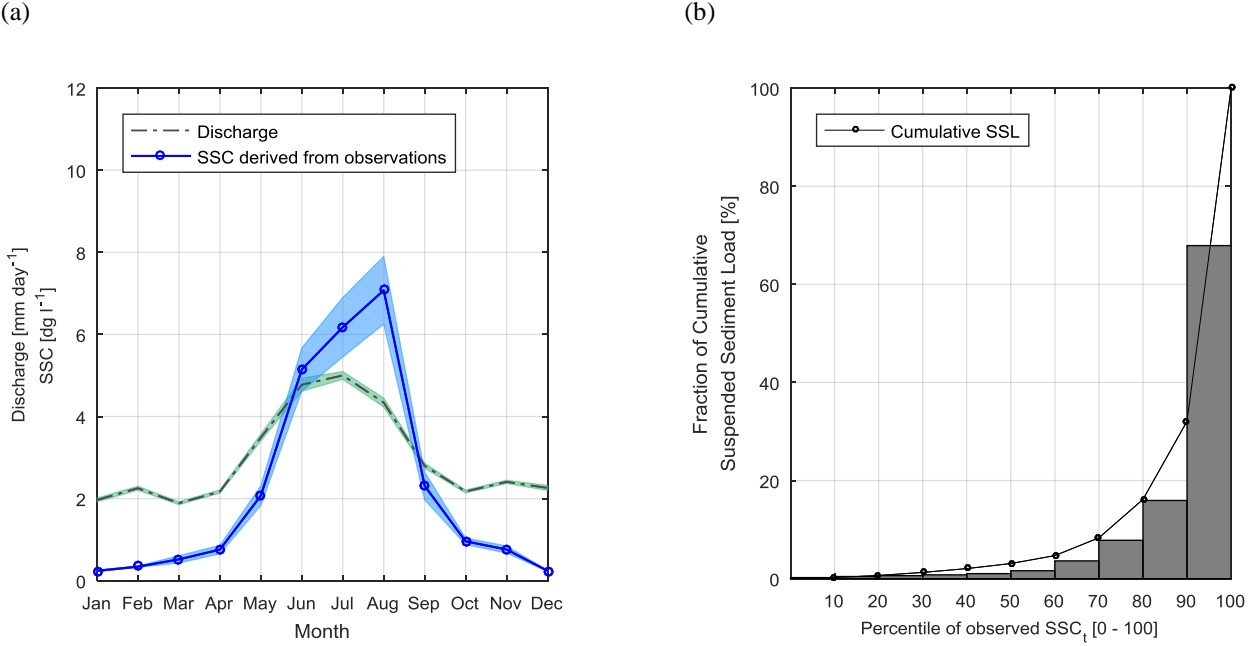

**Figure 2: (a) Mean monthly values of: discharge measured at the outlet of the catchment (dash–dot grey line), SSC$_t$ derived from observations of NTU (solid blue line with circles). Coloured shaded areas represent the range corresponding to ± standard error.**
**Mean values and standard errors are computed over the entire observation period. (b) Cumulative suspended sediment load (SSL) transported at the outlet of the upper Rhone basin during the observation period as function of different percentiles of SSC$_t$ (black line with circles). Bars represent the fraction of the total SSL transported by the different percentiles of SSC$_t$ (e.g.: more than 66% of total SSL is transported with SSC$_t$ > 90$^{th}$ percentile).**



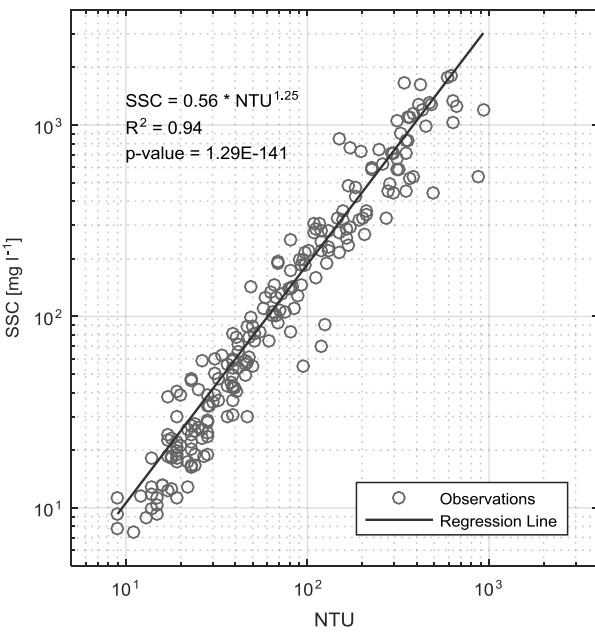

**Figure 3: Scatterplot of NTU and SSC observed simultaneously (i.e. with a maximum lag of 5 minutes) at the outlet of the catchment (grey circles), and calibrated regression line of Eq. 1 (black line).**

## 4 Results

### 4.1 Control of Hydroclimatic Forcing on SSC

The IIS algorithm selects most frequently (56% of the runs) a model with erosive rainfall, icemelt, and snowmelt generated over the unregulated area of the catchment at 1-day lag, $ER_{t-1}$, $IM_{t-1}$, and $SM_{t-1}$, as the most relevant variables to predict mean daily $SSC_t$ (Fig. 4a). We consider only the first 3 selected variables because the cumulative explained variance, expressed as the coefficient of determination $R^2$, is greater than 0.9 (Fig. 4a) and the contribution of additional variables is negligible (the fourth selected variable $ER_{t-2}$ explains roughly 1%). The IIS result is interesting for several reasons. (1) It confirms our hypothesis that erosion and transport processes driven by all 3 hydroclimatic variables ER, IM and SM play a role in determining the suspended sediment dynamics of the Rhone Basin, and likely in most Alpine basins with pluvio–glacio–nival hydrological regimes. (2) It gives an indication of the relative importance of the different processes. In fact, the contribution of each hydroclimatic variable to the overall $R^2$ differs quite significantly. While $ER_{t-1}$ explains almost 75% of the variability of $SSC_t$, the melting components $IM_{t-1}$ and $SM_{t-1}$ are responsible for a much lower fraction of the variance, i.e. 12% and 4% respectively (Fig. 4a). (3) The time lags selected for ER, IM and SM, which represent basin–averaged mean travel times of sediment from source to outlet, including also the time required to produce runoff sufficient to entrain sediment, are equal to 1 day, in agreement with the typical concentration time of the catchment. (4) The most selected model




does not include hydropower releases, HP, (Fig. 4a) indicating that fluxes released from hydropower reservoirs do not play a significant role in determining the variability of the $SSC_t$ signal at the outlet of the basin. When models including hydropower releases are considered (8% of the runs), the first three explanatory variables selected by the IIS algorithm and their explained variance correspond to the ones of the most selected model described above, while hydropower releases are

selected at time lag equal to 0 and represent less than 1.5% of the variability of $SSC_t$ (Fig. 4b). This indicates the characteristic time lag at which the variable HP is considered in the next steps and confirms that it explains only a minor fraction of the variance of SSC. Nevertheless, we include HP in the Hydroclimate Multivariate Rating Curve, HMRC, to assess its contribution to SSC in terms of magnitude and seasonality.

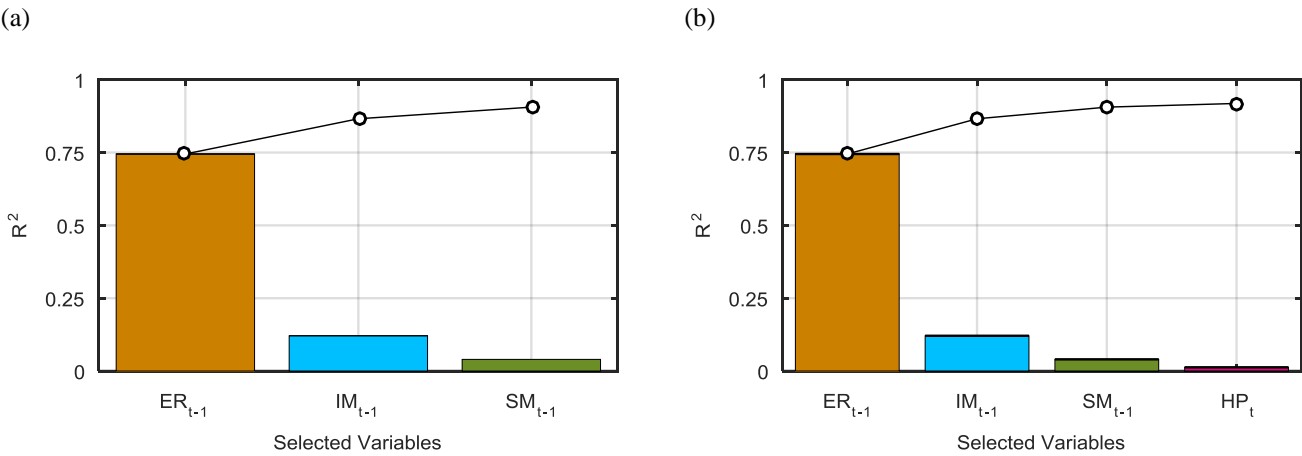

**Figure 4: Results of the IIS algorithm: fraction of the variance of $SSC_t$ explained by the selected explanatory variables, and cumulative explained variance (black line with circles) of (a) the most frequently selected model ($ER_{t-1}$, $IM_{t-1}$, $SM_{t-1}$) and (b) the most frequently selected model including hydropower releases ($ER_{t-1}$, $IM_{t-1}$, $SM_{t-1}$, $HP_t$).**

After the calibration of the parameters (Sect. 2.4), the rating curve based on hydroclimatic variables HMRC and the

traditional RC result respectively in the following forms:

$$SSC_t = \max\ [0.70 \cdot ER_{t-1}^{1.14} + 11.21 \cdot IM_{t-1}^{1.22} + 0.12 \cdot SM_{t-1}^{2.14} - 1.93 \cdot HP_t^{0.47}, 0] \tag{6}$$

$$SSC_t = 0.08 \cdot Q_t^{2.63} \tag{7}$$

where $SSC_t$ is measured in dg l$^{-1}$, the hydroclimatic variables $ER_{t-1}$, $IM_{t-1}$, $SM_{t-1}$ and $HP_t$ are expressed in mm day$^{-1}$, and mean daily discharge $Q_t$ is expressed in mm day$^{-1}$. The values of the parameters of the traditional RC are in agreement with

a previous study on the upper Rhone basin (Loizeau and Dominik. 2000).

Table 1 compares the performances of the HMRC and RC in reproducing mean daily observed $SSC_t$ as measured by the coefficient of determination $R^2$, Nash–Sutcliffe efficiency NSE, and root mean squared error RMSE, over the calibration and validation periods. The HMRC and the RC both show satisfactory performance over the calibration period, e.g. NSE close to 0.6 in both cases, despite the fact that the HMRC does not use observed discharge in the estimation of $SSC_t$. While the





performance of the RC drops in the validation period (e.g. NSE equal to 0.42), the HMRC retains satisfactory performance (e.g. NSE equal to 0.61 and lower RMSE).

**Table 1. Goodness of fit measures for the HMRC and the traditional RC in calibration (left) and validation (right): coefficient of determination ($R^2$), Nash–Sutcliffe efficiency (NSE), root mean squared error (RMSE).**

|  | Calibration 01.05.13 – 30.04.15 | | Validation 01.05.15 – 30.04.17 | |
| --- | --- | --- | --- | --- |
|  | HMRC | RC | HMRC | RC |
| $R^2$ | 0.59 | 0.60 | 0.61 | 0.42 |
| NSE | 0.54 | 0.60 | 0.61 | 0.42 |
| RMSE [dg l$^{-1}$] | 3.25 | 3.02 | 2.66 | 3.23 |

Figure 5 contrasts the HMRC and the RC estimates of mean monthly SSC with SSC derived from observations of NTU from Eq. (1). It is evident that HMRC is better capable to reproduce the seasonal sediment dynamics in all seasons except early spring (February–April). The traditional RC follows discharge seasonality and significantly overestimates SSC in winter and June, generates an early SSC peak, and underestimates SSC in summer (July–September). Perhaps most importantly, mean monthly values of SSC predicted by HMRC in summer, when the amount of sediment transported in suspension is at its highest, are satisfactorily similar to observations.

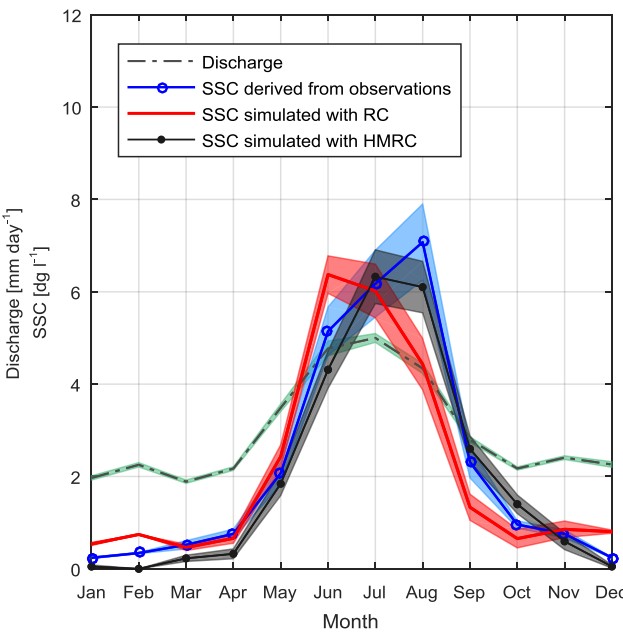

**Figure 5: Mean monthly values of discharge measured at the outlet of the catchment (dash–dot grey line), $SSC_t$ derived from observations of NTU (solid blue line with circles), $SSC_t$ simulated with the traditional RC (solid red line), and with the HMRC (solid black line with dots). Coloured shaded areas represent the range corresponding to ± standard error. Mean values and standard errors are computed over the entire observation period.**





The values of the parameters indicate that IM generates by far the greatest contribution to $SSC_t$ per unit volume of water, followed by ER and SM. The coefficient of the hydropower releases is negative, i.e. water fluxes released from hydropower reservoirs poor in sediment reduce $SSC_t$ in the downstream river by dilution. Over the observation period, IM represents the largest contribution to $SSC_t$ with a mean annual relative contribution equal to almost 40%, followed by ER and SM

5    contributing on average respectively 34% and 26% of $SSC_t$. Figure 6 shows the mean monthly contribution to $SSC_t$ of ER, IM and SM averaged over the observations period. As expected, while IM contributes to $SSC_t$ especially during summer months (July–September), the fraction of $SSC_t$ carried by SM is higher in spring during the snowmelt season (April–June). The effect of erosive rainfall is more evenly distributed throughout the year, and intensified in summer (July–August) when the fraction of the catchment free from snow is at its maximum and rain intensities are high.

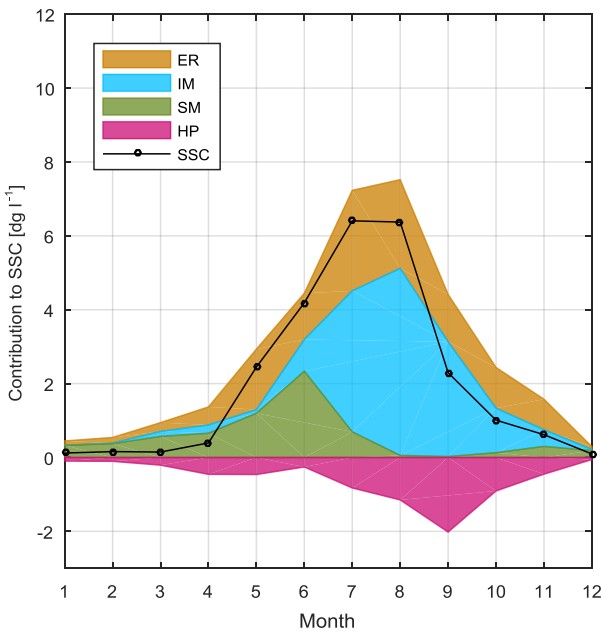

**Figure 6: Mean monthly values of $SSC_t$ computed with HMRC (black line with circles). Coloured areas represent the mean monthly contribution to $SSC_t$ of $ER_{t-1}$, $IM_{t-1}$ and $SM_{t-1}$ and $HP_t$ (dilution) averaged over the observation period.**

## 4.2 Long−term changes in SSC

15    We simulate the HMRC and the traditional RC over the 40–year period 1975–2015 at a daily resolution and compare the two simulations with observations over the same period. We sample only SSC values on days when real twice–a–week observations were taken, to make a fair comparison with observed values which exhibited a jump in 1987. A two–sample two–sided t–test for equality of the mean around this point does reveal a statistically significant jump (5% significance level) only in mean annual SSC values simulated with HMRC and not with RC, if the actual time of the change is known a priori.





Also, if we assume that the time of change is not exactly known, and we compute the probability distribution functions of SSC in two separated periods before and after the observed rise in SSC (namely 1975–1990 and 2000–2015), we conclude that the observations show different distributions in the two periods (Fig. 7a) and that only the HMRC simulation reproduce similar distributions (Fig. 7b) but not the traditional RC (Fig. 7c).

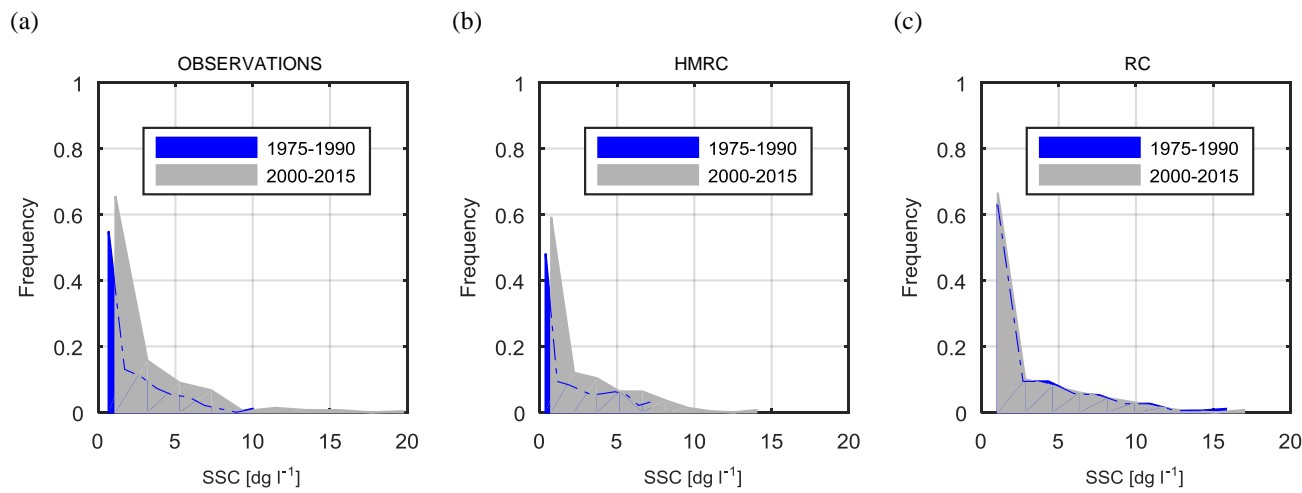

**Figure 7: Empirical probability density functions of mean monthly SSC computed on twice–a–week samples: (a) observed, (b) simulated with HMRC, and (c) with traditional RC for two 15–year periods 1975 – 1990 (blue) and 2000 – 2015 (grey).**

## 5    Discussion

The robustness of the hydroclimatic predictors of SSC in this work depends on the hypothesis that the hydroclimatic variables are independent drivers which activate different sediment sources in Alpine catchments. Indeed the high fraction of the daily $SSC_t$ variance explained by the first three hydroclimatic variables selected by the IIS algorithm, $ER_{t-1}$, $IM_{t-1}$, and $SM_{t-1}$, is in accordance with the physical processes underlying the erosion and sediment transport dynamics in such environments. The higher intensity that characterizes rainfall events in comparison to the melting components is more likely

to generate peaks of SSC during heavy rainfall and floods. In accordance, ER is responsible for a large fraction of the process variability (75%). Indeed, intense rainfall events can detach and mobilize large amounts of sediment (Wischmeier, 1959; 1978; Meusburger et al., 2012). The sharp rise in streamflow, which typically follows a precipitation event, results in an increase in sediment transport capacity that may further entrain sediment previously stored along channels. Precipitation is also one of the main triggering factors of mass wasting events, like landslides and debris flow (e.g. Caine, 1980; Dhakal

and Sidle, 2004; Guzzetti, 2008, Leonarduzzi et al., 2017), in which large quantities of sediment may be instantly released to the river network (e.g. Korup et al., 2004; Bennet et al., 2012). Conversely, the physical processes of icemelt–driven erosion and sediment transport are more gradual and continuous. Similarly, the slow and continuous effect of snowmelt–driven





runoff on hillslope and channel erosion contributes to the seasonal pattern of SSC and plays a secondary role in explaining its daily variability and peaks in SSC. Interestingly, hydropower releases do not influence significantly the variance of SSC, despite the fact that the Rhone Basin is heavily regulated by hydropower reservoirs. This is most likely related to the fact that water fluxes downstream of Alpine hydropower dams have lower concentrations of suspended sediment compared to fluxes

entering the reservoirs, due to sediment trapping in the reservoirs (Loizeau and Dominik. 2000; Anselmetti et al., 2007). This is in agreement with results of the sediment fingerprinting analysis recently performed in the catchment, which suggests the underrepresentation of sediments originated in the most highly regulated lithological unit (Stutenbecker et al., 2017). This is also indicated by the negative coefficient of HP, which suggests that hydropower releases dilute suspended sediment in the HMRC model and therefore leads to a reduction of $SSC_t$ compared to natural flow. It should also be noted that the effect of

hydropower reservoirs on sediment storage is grain size dependent (e.g. Anselmetti et al., 2007) and may be substantially different for coarser grains transported as bedload.

We calibrate and validate a rating curve based on the hydroclimatic variables selected by the IIS algorithm and hydropower releases (HMRC) and a traditional rating curve (RC) based on discharge only. While both the HMRC and the traditional RC show similar performance in calibration, the HMRC by taking into account the physical processes which govern SSC in a

more direct way, performs better in validation, simulates more accurately the seasonal pattern of SSC, especially in summer when melting of snow and ice are active and a large fraction of the catchment is snow free and subject to erosion by rainfall. The traditional RC overestimates SSC in winter because it relies on streamflow only and does not account for the low concentration of sediment coming from hydropower reservoirs.

On the basis of the HMRC parameters, we find that although ER is responsible for the peaks of SSC and therefore,

contributes the most to the variance of SSC, IM fluxes generate the highest SSC per unit volume of water. This is in agreement with the fact that meltwater originated in glaciated areas is characterized by very high sediment concentrations (Gurnell et al., 1996; Lawler et al., 1992). For a catchment significantly glacierized such as the upper Rhone basin (roughly 10% of the surface is covered by glaciers) this implies also that among the hydroclimatic variables, IM represents the greatest contribution to SSC and suspended sediment yield from this Alpine catchment (as shown in Fig. 6). This supports

the findings of Costa et al. (2017) where the authors show that the increase in SSC observed at the outlet of the Rhone Basin in mid–1980s is most likely due to a significant rise in icemelt fluxes due to the enhanced glacier retreat associated with warmer temperatures. In concurrence with increasing icemelt, the mean annual SSC at the outlet of the catchment generated by IM, as simulated by the HMRC, increases after mid-1980s (Fig. 8). This explains why the HMRC is capable of simulating the observed shift in SSC, although the simulation resembles more a gradual increase than a sudden jump. The results show

that a more process based rating curve accounting for the different hydroclimatic forcing can not only separate the relative effects of the different forcings on SSC, but also explain climate–driven changes in suspended sediment dynamics, which is not possible by adopting a traditional rating curve based on discharge alone.





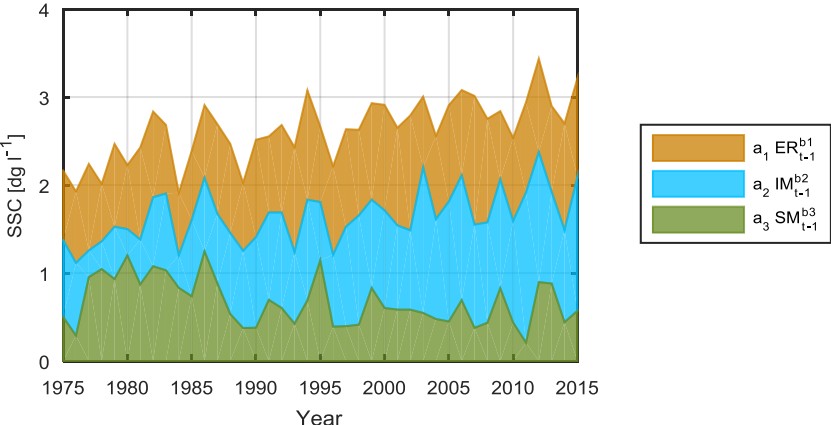

**Figure 8: Mean annual contribution to SSC of $ER_{t-1}$, $IM_{t-1}$ and $SM_{t-1}$ simulated with the HMRC.**

## 6 Conclusions

In this paper, we analyse how hydroclimatic factors influence suspended sediment concentration (SSC) in Alpine catchments
by differentiating among the potential contributions of erosional and transport processes typical of Alpine environments,
driven by (1) erosive rainfall (ER) defined as liquid precipitation over snow free surfaces, (2) icemelt (IM), and (3) snowmelt
(SM). For regulated catchments, we include the potential effect of hydropower by considering the contribution to SSC of
fluxes released from reservoirs due to hydropower operations (HP). We obtained the hydroclimatic variables ER, SM, IM by
using a conceptual spatially distributed model of snow accumulation, snow and ice melt driven by precipitation and
temperature at a daily resolution and we computed HP via a unique virtual reservoir which was operated on the basis of a
target volume function, which is aimed at reproducing the cumulated effect of the historical operations of the several
hydropower facilities. We then used the Iterative Input Selection (IIS) algorithm to select the variables that play a significant
role in predicting SSC and to quantify their relative importance and predictive power in simulating observed changes in SSC
in the Rhone Basin over a period of 40 years. We tested our approach on the upper Rhone basin in Switzerland. Our main
findings can be summarized as follows.

(1) The three hydroclimatic processes ER, IM, and SM are significant predictors of mean daily SSC at the outlet of the upper
Rhone basin, explaining respectively 75%, 12% and 4 % of the total observed variance; hydropower releases HP do not play
a significant role in defining the variance of SSC most likely because fluxes released from reservoirs are poor in sediment
due to sediment trapping. The characteristic time lag of 1 day for the ER, IM and SM fluxes, representing the time necessary
to produce sufficient runoff and to entrain and transport sediment from a given location in the catchment to the outlet, are in
agreement with typical concentration times of the catchment; conversely for HP the time lag is lower than one day.

(2) Although ER is responsible for the greatest fraction of the variability of SSC at a daily basis, coefficients of the HMRC
indicate that IM generates the greatest contribution to SSC per unit of water volume and contributes the most in terms of



mean annual sediment yield. This is in agreement with the high suspended sediment concentration that characterizes icemelt fluxes and with finding of previous studies that indicate the increase in icemelt as most plausible explanation of changes in suspended sediment dynamics in the catchment (Costa et al., 2014).

(3) The HMRC is capable of reproducing the pattern of SSC even though it does not include discharge as an input variable.
Although the HMRC and traditional discharge based RC perform similarly in simulating observed SSC over the calibration period, the HMRC performs better than the traditional RC in validation at the daily scale, and in capturing seasonality, especially in summer when SSC are highest. This is particularly relevant because more than 66% of the total suspended sediment load reaching the outlet of the upper Rhone basin in the observation period is transported by SSC values larger than the 90[th] percentile.

(4) With the HMRC approach we are able to reproduce changes in SSC in the past 40 years that have occurred in the catchment due to a temperature change, and we can demonstrate that the shift in SSC is most likely due to the increase in icemelt fluxes.

In summary, our approach provides an insight on how hydroclimatic variables control SSC dynamics in Alpine catchments, and the results suggest that a more process and data informed approach in predicting suspended sediment concentrations, which accounts for sediment sources and transport processes driven by erosive rainfall, snowmelt and icemelt, instead of only discharge, allows to analyze climate–induced changes in sediment dynamics. Although these results are specific for the upper Rhone basin only, the approach is general and may be employed in other Alpine catchments with pluvio–glacio–nival hydrological regimes where sufficient data are available.

**Author contribution**

A. Costa, D. Anghileri and P. Molnar designed the methodology. A. Costa and D. Anghileri developed the code and carried out simulations and computations. All co–authors contributed to the manuscript. The authors declare that they have no conflict of interest.

**Acknowledgements**

We thank the Federal Office of the Environment (FOEN) for providing discharge, suspended sediment concentration and turbidity data. We also thank Alessandro Grasso (FOEN) for the explanation on the SSC and turbidity measurement procedures. This research was supported by the Swiss National Science Foundation Sinergia grant 147689 (SEDFATE). Daniela Anghileri was supported by the Swiss Competence Centre on Energy – Supply of Energy (SCCER–SoE).

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
