# Peer review of "Hydroclimatic control on suspended sediment dynamics of a regulated Alpine catchment: a conceptual approach"

_Hydrology and Earth System Sciences, 2018_

## Editor Comment (EC1) · Prof. Mikos (Editor) · 11 Jan 2018

Dear Authors of the re-submitted manuscript,

during the Interactive Discussion of the original manuscript hess-2017-419:

https://www.hydrol-earth-syst-sci-discuss.net/hess-2017-419/

three Referee Comments (RCs) have been published. Before closing the Interactive Discussion, you have answered to the first two reviewers by your Authors Comments (AC), but not to the third Referee Tammo Steenhuis, who published his review (RC3) on September 6, 2017, during the Interactive Discussion.

[Figure]

In order to guide the Interactive Discussion of this re-submission hess-2018-0005, please, publish as soon as possible firstly your reply (AC) to the Referee #3 Comments of the original submission.

Kind regards,

Matjaž Mikoš Handling Editor

---

## Author Comment (AC1) · 15 Jan 2018

Reply to Referee 3 – T. Steenhuis

The original paper was rejected, and we reworked the analysis and rewrote the paper to focus it on the contribution that we believe our work has to the understanding of hydroclimatic signals in suspended sediment concentration in the Alpine catchment we studied. Many of the comments of the referee were used to motivate our reanalysis and rewriting. In this post-rejection response we summarize how the new resubmitted paper addresses the main comments/concerns of the referee.

[Figure]

(1) REF: A great number of abbreviations are introduced, making the article almost impossible to read. Abbreviations are not explained in the figures. Figures should stand alone and abbreviations should be mentioned. Headings of sections have abbreviations. Just too much jargon. There should be some previous research done on Alpine sediment fluxes and processes. The authors do not mention any of these research studies in the following statement. . .

RESPONSE: We have rewritten the paper, shortened and focussed it. We believe that we have cited the most relevant Alpine research on suspended sediment dynamics. Although, as the referee is well aware, there is an extraordinary amount of work published on this topic, so we have had to be selective. If we ommited any relevant work we are sorry for that and will be happy to receive pointers to concrete papers.
* * *
(2) REF: I doubt very much that the signal of raindrop impact is preserved somewhere hundreds of kilometers down. Pick up of sediment in rills of plowed soils and deposition afterwards could completely overwhelm the raindrop impact signal (Moges et al., 2016). Moreover in mountainous environment saturation excess rainfall dominates and total rainfall explains better the runoff amounts and sediment concentrations than the loads than the intensity (Tilahun et al., 2013, 2015). Unlike what the authors write in their manuscript it is the total rainfall in Guzman et al (2013) that is related to the sediment concentrations in the Ethiopia highlands and not the rainfall intensity (page 3 around line 10). I do not know how the Alpine environments are different from the Ethiopia highlands, but that it is the task of the authors to research the processes that are really occurring in the Alpine watersheds.

RESPONSE: We did research the processes in Alpine watersheds. Indeed rainfall intensity is an important factor for rainfall erosivity (Meusburger et al., 2012), while erodibility of the surface is strongly related to lithology, soil texture, rock fraction, etc. (Prasuhn et al., 2013). For fine sediment, which is what we are studying, glacier retreat and the connectivity of the colluvial and supraglacial sediment to the fluvial system are

additional key factors (Lane et al., 2017). In contrast to Ethiopia there is practically no intensive agriculture in the Rhone Basin outside the floodplain (vineyards). Hillslope sediment supply is by ephemeral gullies and steep torrents, which are active during rain on snow-free surfaces. This does not rule out that there is a correlation between cumulated total rainfall amounts and sediment concentrations as the referee states.
* * *
(3) REF: It is interesting that in the response to the reviewer 2 the authors write "Our empirical model partially accounts for the effect of hydropower operations on SSC magnitude and timing because we calibrated the parameters of the PBRC using the observed time series of SSC at the outlet of the basin, which are impacted by the hydropower operations. Therefore, the coefficients (a1, a2, a3), the exponents (b1, b2, b3) as well as the time lags specific of each hydro-climatic variable (l1, l2, l3) include the impacts of reservoirs and hydropower operations (e.g. delays in sediment transfer are accounted for in the time lags chosen by the IIS algorithm). Even though our approach is relatively simple and cannot capture all the complexities of the sediment..." This contradicts the earlier explanation above about the processes in the watershed. In other words the authors present a model fitting routine that is inspired by some of kind reality but reality has ultimately very little to do with the explanation of the results. For example, what would happen to the sediment concentration when the reservoir operation changes? This cannot be simulated by the sediment rating curve model.

RESPONSE: We do think that reality has something to do with the explanation of the results. However, this referee comment has allowed us to reflect on the analysis and motivated us to redo the work. To recap, our aim was not to come up with a new "better" suspended sediment rating curve, which perhaps was the thinking of the referee. Our aim was to use sediment and climate data to explore the predictive power in suspended sediment concentrations in a large regulated Alpine catchment coming from hydroclimatic forcing variables, and not from discharge. These variables were chosen in such a way that they activate different sediment sources in time: e.g. erosive rainfall on

snow free surfaces and snowmelt are representative for hillslope and channel erosion, while icemelt is representative for the supply of sediment generated by glacial erosion. We are using data in the post-dam period, so hydropower operations are explicitly included in the data and cannot be removed. But it is true that hydropower operations were not explored in the original paper. To fix this, in the revised version we have decided to elaborate a fourth variable – hydropower releases – which are now estimated from a typical target level operation of reservoirs in the system. The results show that indeed hydropower releases have a lower predicitive power than all three hydroclimatic forcing variables, and in fact that the effect on sediment concentration is negative, i.e. releases of sediment poorer water from reservoirs is lowering the mean sediment concentration in the downstream receiving waters. This is logical and intuitive. The message we try to convey with our new work is that hydroclimatic forcing signals can be seen in suspended sediment concentrations even in a rather heavily regulated Alpine catchment, such as the Rhone. There is evidence for this fact at the small subbasin scale (e.g. Lane et al., 2017; Bakker et al., 2018) but the fact that climate sensitivity can be seen even at the large Rhone Basin scale was not fully appreciated to–date.
* * *
(4) REF: It is well known from the equifinality approach of (Beven and Freer, 2001) that many different combinations of parameters give a best fit for the signal at the outlet. The authors found this best fits using 6 parameters. There could be a range of parameter sets that give the same result. Moreover the model chosen might not be the best. It should be addressed how optimum is the fitting parameter set. This review is not to indicate that the manuscript cannot be published ultimately. However any fancy explanations about what happens in the watershed based on the fitting of the sediment concentrations seems out of place unless it can be shown based on experimental evidence that the parameters in the watershed affect the concentration at the outlet.

RESPONSE: This is a data-based study. We are simply searching for relations between variables (in a multi-variate setting), either observed or simulated with validated and tested approaches. In this search it may of course happen that several combinations of variables give similar result. It is in fact a feature of the input variable selection algorithm that the "best" combinations are sought starting from different combinations, and so their rubustness is quantified. It is explained in our resubmitted work how the algorithm works, and we stress the fact that the selection of the "best" variables is robust. Likewise, the optimization procedure applied to calibrate the 6 parameters of the equation is repeated 100 times, starting from randomly generated initial values to reduce the risk of detecting sub–optimal parameter configurations. We do not have experimental evidence and of course we do not have proof like that a physically-based model would provide, but we are working on this possibility for the future.
* * *
(5) REF: Simply in my opinion, the manuscript is about a model with six parameters that is fitting the output of some kind of simple hydrological model to very detailed measurements of daily sediment concentration. The authors should be realistic what can be done with the model and where it can be used for. For example how is the management of the hydropower dams been changed over the years and can that be the reason that the sediment concentration are changing?

RESPONSE: This statement and question was driving most of our revision and preparation of the resubmission. We agree that the effect of hydropower dams was not explicitly included in our work, and we decided to change this by adding a hydropower term to the predictive equation and comparing the predictive power of hydroclimatic forcing with that of hydropower (which results in lower predicitive power than that of hydroclimatic variables, as explained above). The use of the model is not to replace discharge-based rating curves. Furthermore it is laborious to derive all the hydroclimatic variables over any watershed by running a model or by interpolation. The use of the model is that it allows for the relative predictive power of the individual hydroclimatic variables to be quantified. This not only explains why highest suspended sediment

concentrations are still found after heavy rain, but it also gives an explanation for the longer-term changes in concentrations driven by warming and subsequent increases in icemelt. Quantifying these connections with discharge only is not possible.

---

## Referee Comment (RC1) · Anonymous Referee #1 · 18 Feb 2018

Authors have prepared a revised version of the manuscript where they have put more focus on the investigation of the hydroclimatic controls on the suspended sediment dynamics. Authors have considered my previous comments (https://www.hydrol-earth-syst-sci-discuss.net/hess-2017-419/) and mostly incorporated them in the revised version of the manuscript. Thus, I suggest to accept the resubmitted/revised paper.

---

## Referee Comment (RC2) · T. Steenhuis (Referee) · 17 Apr 2018

T. Steenhuis (Referee)

tss1@cornell.edu

The improvements to the paper are impressive and the response clearly convinces me that the paper should be published.

I do not think that the authors solved the equifinality problem completely because the many processes occurring in the watershed integrate to one signal at the outlet. The parameter combination found might be indeed the best fitting set, but that does mean that there are no other sets that almost fit just as well. However, equifinality is a problem for all modelers and it would not be fair to draw a line in the sand here. Moreover, as long as the fitted parameter combination makes physically sense, it is likely right. I am

glad to the authors are working on experimental evidence for the future.

Thank you for making the changes. My apologies for the long delay in responding. I should have done it much sooner

Tammo Steenhuis

---

## Author Comment (AC2) · 23 May 2018

We thank Referee #1 for her/his comment and for her/his previous review which allowed us to improve the manuscript.

---

## Author Comment (AC3) · 23 May 2018

Reply to Referee #2 - Tammo Steenhuis

We would like to thank Referee #2 for his previous review which motivated us to improve both the analysis and the manuscript. We thank Referee #2 for raising the issue of equifinality. We are aware of this problem and we agree with the reviewer that equifinality is a potential issue in any modeling based study. We will clarify in the final version of the manuscript that we addressed this issue by repeating several times both the input variable selection algorithm and the model parameter optimization and, although the equifinality problem may be not fully overcome, the final parameter set is reason-

able on the basis of the physical interpretations of the results which are included in the discussion.

—————————————————

---

## Author Response (AR1)

**Author' response [paper hess-2018-5]**

**Hydroclimatic control on suspended sediment dynamics of a regulated Alpine catchment: a conceptual approach**

Anna Costa, Daniela Anghileri, Peter Molnar

We thank the Editor and the reviewers for their comments. We have implemented the changes that the Editor suggested and we have added comments to clarify how we addressed the issue of equifinality, in agreement with our reply to T. Steenhuis (Reviewer #2).